# Indications for bi-cruciate retaining total knee replacement: An international survey of 346 knee surgeons

Diarmuid De Faoite[1]*, Christian Ries[2], Michelle Foster[3], Christoph Kolja Boese[4,5]

**1** Smith + Nephew, Baar, Switzerland, **2** Department of Orthopedics, University Medical Center Hamburg-Eppendorf, Hamburg, Germany, **3** Smith + Nephew, Hull, United Kingdom, **4** Department of Orthopaedic and Trauma Surgery, University Hospital of Cologne, Cologne, Germany, **5** Smith + Nephew GmbH, Hamburg, Germany

* diarmuid.defaoite@smith-nephew.com

**Data Availability Statement:** All relevant data are within the manuscript and its Supporting Information files.

**Funding:** The study was funded by Smith + Nephew as an investigator initiated study (IIS 585).

## Abstract

There is limited evidence on total knee arthroplasty (TKA) that retains the anterior and the posterior cruciate ligaments. Bi-cruciate retaining (BCR) TKA is considered to show improved clinical function and kinematics. This survey aimed to (1) identify interest in and acceptance of BCR TKA surgery and (2) to capture the range of indications for BCR in the opinion of practicing knee surgeons. 346 surgeons with experience in TKA surgery completed a bi-lingual online survey. Demographics, arthroplasty experience as well as acceptance of and indications for BCR TKA were collected. 53 surgeons were experienced in BCR TKA and 225 would consider implanting it. A mean of 19.5% of TKA patients were considered eligible for BCR TKA. 56.3% were thought to have intact ACL at the time of TKA surgery. Surgeons were not likely to perform BCR TKA in patients with inflammatory arthritis, aged over 80, BMI above 34.9 kg/m$^2$, a varus or valgus deformity of more than 10˚ and flexion contractures of more than 10˚. There is strong interest among orthopedic surgeons to perform BCR TKA and the percentage of potentially eligible patients is high. Significant joint deformity is a limitation of BCR TKA, while age and high BMI are less relevant. BCR TKA experienced surgeons are less restrictive.

## Introduction

Most total knee arthroplasty (TKA) techniques involve sacrificing the anterior cruciate ligament or both cruciate ligaments.[1, 2] Few TKA systems allow for the preservation of both cruciate ligaments.[2, 3] TKA that retains the posterior cruciate ligament (PCL) as well as the anterior cruciate ligament (ACL) is referred to as bi-cruciate retaining (BCR) TKA.[3–5]

Functionally intact cruciate ligaments are required for BCR TKA. Studies show that up to 78% of ACLs are intact at the time of the TKA.[6, 7] However, BCR TKAs are seldom implanted and few implant systems are available.[3] While it is assumed that retaining the ACL may result in more natural knee kinematics contributing to higher patient satisfaction,[2, 3, 8] there are concerns about technical difficulties, complication risks, long term survival and

The funder provided support in the form of coverage of costs associated with the survey (fee for online services by SurveyMonkey), coverage of publication fees (if applicable) and salaries for authors CKB, DDF and MS, but did not have any additional role in the study design, data collection and analysis, decision to publish, or preparation of the manuscript. The specific roles of these authors are articulated in the 'author contributions' section.

**Competing interests:** "This does not alter our adherence to PLOS ONE policies on sharing data and materials." "MS and CKB may be eligible to stock / stock options of Smith & Nephew PLC." "The authors do not declare any other competing interest."

functional outcomes.[3, 9, 10] A systematic literature review of the available BCR TKA clinical evidence revealed outcomes comparable to other TKA designs.[3] However, there is a lack of knowledge on the relevant indications for BCR TKA, and most importantly, the limitations to indications.

This survey aimed to (1) identify interest in and acceptance of BCR surgery and (2) to capture the range of indications for BCR in the opinion of practicing knee surgeons.

## Materials and methods

A bi-lingual (German and English) questionnaire was developed. The questionnaire was published online through SurveyMonkey™ (San Mateo, CA, US). Questions posed included information about demographics, arthroplasty experience as well as acceptance of and indications for BCR TKA.

From October 2018 to March 2019, surgeons were contacted personally by the authors or by e-mail using the internal GDPR-compliant distribution list of healthcare professionals. All communication was aligned with global and local laws, as well as privacy and data protection regulations. The ethics commission (IRB) granted a waiver as long as no personal identifiers were collected as part of the study (Ethikkommission der Ärztekammer, Hamburg, Germany). No personal identifiers were collected. All participants electronically agreed to the terms and conditions.

### Survey design

The survey design was informed by unstructured interviews with senior knee surgeons (n = 11) and a systematic literature review on BCR TKA. The draft survey was circulated to a team of three in-house orthopedic surgeons and the clinical, scientific and medical affairs team for content development. Repeated circulation of modified survey versions was performed until consensus was found. Due to the use of skip logic in the questionnaire, not all questions were answered by each respondent.

The survey was available in two languages (English, German). Native speakers of both languages performed forward and backward translations until consensus was found. The survey consisted of 39 questions, the final being an open ended comment field. The complete questionnaires in both languages are available as supplemental material (S1 File).

Inclusion criteria required respondents to be a) a medical doctor (MD) with experience in orthopedic surgery and b) with experience with TKA. Inexperience with TKA or any non-doctors were therefore excluded from the survey.

Incomplete surveys were eliminated from the analysis per protocol. Contradictory answers to specific questions of completed surveys were ignored. Free text comments were stratified into specific answer categories for analysis where appropriate.

### Stratification

**Answers given as percentages.** Stratification into groups of 20% increments was performed (e.g. 1–20%, 21–40%, etc.).

**Age.** The upper age limitations considered appropriate for BCR TKA indication were stratified into groups of <45 years (young), 45–64 years (middle-aged), 65–79 years (aged) and 80 years or older using the Medical Subject Headings (MeSH) classification.[11]

**BMI.** The upper body mass index (BMI) limitations considered appropriate for BCR TKA indication were stratified into groups using the World Health Organization (WHO) classification.[12] The categories used were underweight ($<18.5$ kg/m$^2$), normal weight (18.5–24.9 kg/

m$^2$), pre-obesity (25–29.9 kg/m$^2$), obesity class I (30–34.9 kg/m$^2$), obesity class II (35–39.9 kg/m$^2$) and obesity class III (40+ kg/m$^2$).

**Statistical analysis.** The results from the German and English language surveys were pooled into one dataset. A senior biostatistician (MF) conducted the statistical analysis. Data output from SurveyMonkey was exported as a Microsoft Excel file (Microsoft Office Professional Plus 2016, Microsoft Corp., Redmond, WA, USA). The data analysis for this paper was generated using SAS software, (Version 9.4., SAS Institute Inc., Cary, NS, USA).

Continuous variables were summarized with the following summary statistics: number of observations, mean, median, standard deviation, minimum and maximum values. Categorical variables were summarized as frequencies and percentages. Histograms and box-plots were used to visualize data. Unless otherwise stated, all significance tests and hypothesis testing were two-sided, performed at the 5% significance level. All variables were tested for normality using a Shapiro-Wilks normality test and non-parametric tests were applied where appropriate. Where statistical significance was achieved, post-hoc tests were used and Bonferroni corrections were applied to adjust for the number of all possible pairs. Spearman's Rank correlation coefficients were calculated for correlation analysis.

# Results

## Baseline characteristics of respondents

346 completed surveys were collected and analyzed. 82 (23.7%) were from the German language survey and 264 (76.3%) from the English language survey. Baseline demographic characteristics of the respondents are displayed in **Table 1**. Information on arthroplasty experience is presented in **S1 File**.

## Key outcomes

53 surgeons (15.3%) had operative experience with BCR implants. The most commonly used BCR implants were Journey$^{TM}$ II XR (Smith + Nephew plc., Watford, UK) (n = 38; 71.7%), Vanguard XP$^{TM}$ (Zimmer Biomet, Warsaw, Indiana, USA) (n = 15; 28.3%) and others (n = 19; 35.8%).

225 (65.0%) of surgeons would consider implanting a BCR total knee prosthesis; 65 (18.8%) were not sure; and 56 (16.2%) would not.

135 surgeons (39.0%) always sacrifice the PCL during TKA surgery, 140 surgeons (40.5%) sometimes sacrifice the PCL and 34 surgeons (9.8%) never sacrifice the PCL. 37 respondents (10.7%) did not record a response to this question.

**Table 1. Baseline characteristics.**

|  | Answer | N (%) |
|---|---|---|
| **Sex** | Male | 332 (96.0%) |
|  | Female | 12 (3.4%) |
|  | Prefer not to say | 2 (0.6%) |
| **Position** | Attending Physician | 279 (80.7%) |
|  | Fellow | 26 (7.6%) |
|  | Resident | 21 (6.0%) |
|  | Other | 20 (5.8%) |

German job titles have been amalgamated. Attending physician includes *Chefarzt* and *Oberarzt*. *Facharzt* is Fellow. *Weiterbildungsassistent* is resident.

**Table 2. Percentage of suitability for BCR.**

| Percentage of patients suitable for BCR implant | N (%) | Do you sacrifice the PCL during TKA procedures?[1] | | | | Would you consider implanting a BCR total knee prosthesis? | | | |
|---|---|---|---|---|---|---|---|---|---|
| | | Always | Some-times | Never | Total | Yes | No | I do not know | Total |
| None | 61 (17.6%) | 32 | 18 | 10 | **60** | 17 | 28 | 16 | **61** |
| 1–20% | 173 (50.0%) | 65 | 73 | 16 | **154** | 115 | 24 | 34 | **173** |
| 21–40% | 58 (16.8%) | 23 | 25 | 3 | **51** | 45 | 2 | 11 | **58** |
| 41–60% | 37 (10.7%) | 9 | 18 | 3 | **30** | 34 | 0 | 3 | **37** |
| 61–80% | 17 (4.9%) | 6 | 6 | 2 | **14** | 14 | 2 | 1 | **17** |
| 81–100% | 0 (0%) | 0 | 0 | 0 | **0** | 0 | 0 | 0 | **0** |
| **Total** | 346 | 135 | 140 | 34 | 309 | 225 | 56 | 65 | 346 |

[1] 37 missing responses.

A mean of 19.5% (median 13, SD 19.99; range 0–80) of patients indicated for a knee replacement were deemed suitable for BCR (Table 2).

There was a statistically significant difference in the percentage of patients suitable for BCR TKA implantation between surgeons' preference of sacrificing the PCL (i.e. always, sometimes, never) during TKA procedures (p = 0.0095). Surgeons sometimes sacrificing the PCL were most likely to indicate BCR TKA ("sometimes" vs. "always": p = 0.0327; "sometimes" vs. "never": p = 0.0516).

Surgeons with experience with partial knee arthroplasty (PKA) were more likely to indicate BCR TKA compared to those without PKA experience (p = 0.0041). The willingness to indicate BCR TKA showed significant differences between surgeons' experiences with different types of PKA (Table 3).

Surgeons reported the percentage of patients presenting with a functionally intact ACL at the time of TKA surgery to be 56.3% (median 60, SD 23.01; range 0–100) (S2 File and S3 File). Participants would apply MRI (67%, n = 232) or a pre-operative clinical function test (90%, n = 311) of the ACL to determine the ACL status (S4 File). 136 participants (39%) would indicate BCR TKA in ACL with signs of degeneration (S4 File).

The BCR indication spectrum overlapped with indication for PKA, TKA or a treatment gap between PKA and TKA (Fig 1).

## Indications

**Age.** 167 (48.3%) surgeons considered age a relevant indication factor. 153 (91.6%) defined an upper age limit. The mean limit was 68.0 years (n = 153; median 70; SD 9.82; range 36–99) (Table 4).

**Table 3. Experience in partial knee arthroplasty.**

| | Percentage of patients indicated for a knee replacement suitable for BCR | | | | | | | | | | | |
|---|---|---|---|---|---|---|---|---|---|---|---|---|
| | Experience with PKA | | | | | No experience with PKA | | | | | | |
| | N | Mean | Median | SD | Range | N | Mean | Median | SD | Range | Z score | p-value[1] |
| All Partial Knees | 274 | 20.5 | 15 | 19.87 | 0–80 | 72 | 15.5 | 6 | 20.07 | 0–80 | -2.871 | 0.0041 |
| UKA | 240 | 20.1 | 15 | 19.48 | 0–80 | 106 | 18.0 | 10 | 21.13 | 0–80 | -1.781 | 0.2250[2] |
| PJF | 114 | 20.2 | 15 | 19.54 | 0–80 | 232 | 19.1 | 10 | 20.24 | 0–80 | 1.007 | 0.9417[2] |
| Bicompartmental PKA | 83 | 25.5 | 20 | 20.65 | 0–80 | 263 | 17.6 | 10 | 19.43 | 0–80 | 3.919 | 0.0003[2] |

[1] Wilcoxon Signed Rank test

[2] Bonferroni correction applied

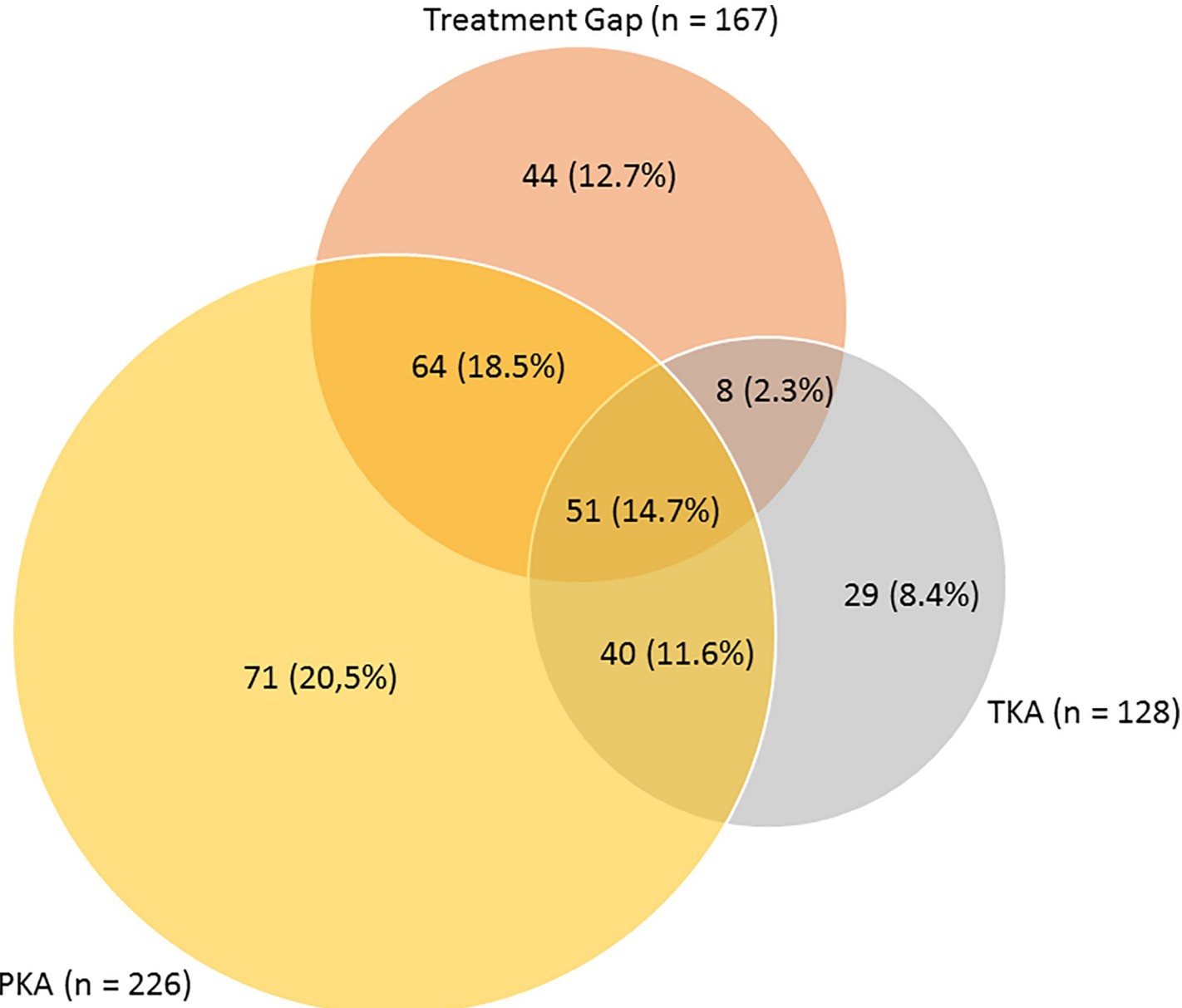

**Fig 1. Multiple choice selection of indications for BCR TKA.** Options were same indication as (1) partial knee arthroplasty (PKA), (2) total knee arthroplasty (TKA) or a treatment gap between the two.

**BMI.** 179 (51.7%) surgeons found BMI to be a relevant factor for BCR TKA indication. Of these, 177 (98.9%) defined an upper limit. The mean limit was 32.6 kg/m$^2$ (n = 177; median 34; SD 4.26; range 25–40) (**Table 5**).

**Contracture.** 278 (80.3%) surgeons considered flexion contracture to be a relevant factor for BCR TKA indication. Contracture was reported as maximum acceptable degrees. 7 (2.5%) respondents selected a maximum of 90˚ defined as "no limitation". Excluding answers without limits, the mean limit was 11.5˚ (n = 271; median 10; SD 14.64; range 0–80) (**Fig 2**).

**Deformity.** 268 (77.5%) surgeons deemed deformities a relevant factor for BCR TKA indication. Of these, 251 (93.7%) would accept varus deformities with a mean of 11.9˚ (n = 251; median 10; SD 5.91; range 5–35). 236 (88.0%) respondents would accept valgus

**Table 4. Indication age.** Stratified by MeSH groups.

| | Upper limit | |
|---|---|---|
| Age group | N | % |
| < 45 | 2 | 1.3% |
| Middle aged (45–64) | 49 | 32.0% |
| Aged (65–79) | 82 | 53.6% |
| 80 or more | 20 | 13.0% |
| Total | 153 | 100% |

179 surgeons did not answer this question.

deformities with a mean of 10.6˚ (n = 236; median 10; SD 5.16; range 5–35). 254 (94.8%) surgeons answered the acceptable posterior slope was limited to a mean of 9.4˚ (n = 254; median 10; SD 4.44; range 5–40) (**Table 6**).

Willingness or unwillingness to indicate BCR TKA did not influence the distribution of indication ranges. Surgeons' preference for PCL sacrifice only affected upper BMI as limitations to indications (p = 0.0274). P-values are provided for every indication category in **Tables 7 and 8.**

## Discussion

This survey collected the opinions of practicing knee surgeons on BCR TKA. The primary aim was characterization of interest in and acceptance of BCR surgery. Secondary goals were descriptions of perceived ranges and limitations of indications for BCR TKA. Overall, 346 surgeons with experience in knee arthroplasty completed the survey. The respondents covered a wide range of experience including high, medium and low volume knee arthroplasty. Only 3.4% were female and 88.3% were attending physicians or fellows with extensive experience. Notably, a relevant number of surgeons also had experience with PKA but only 15.3% had previously performed a BCR TKA.

There was high interest in performing BCR TKA. 65.0% of the participants would consider implanting a BCR TKA and another 18.8% were not sure. Only a few surgeons would not consider it. Additionally, 222 (64.2%) surgeons said there is a need for BCR TKA.

Despite this strong interest in BCR TKA, there were concerns regarding the technical difficulty of the procedure. While this is partially supported by existing literature,[13] there is no clear indication of more complications.[3] Although Christensen et al. found higher revision

**Table 5. Indication BMI.** Stratification by WHO categories.

| | Upper limit | |
|---|---|---|
| BMI (kg/m$^2$) | N | % |
| Less than 18.5 | 0 | 0% |
| 18.5–24.9 | 0 | 0% |
| 25–29.9 | 33 | 18.7% |
| 30–34.9 | 62 | 35.0% |
| 35–39.9 | 63 | 35.6% |
| 40 + | 19 | 10.8% |
| Total | 177 | 100% |

167 surgeons did not answer this question.

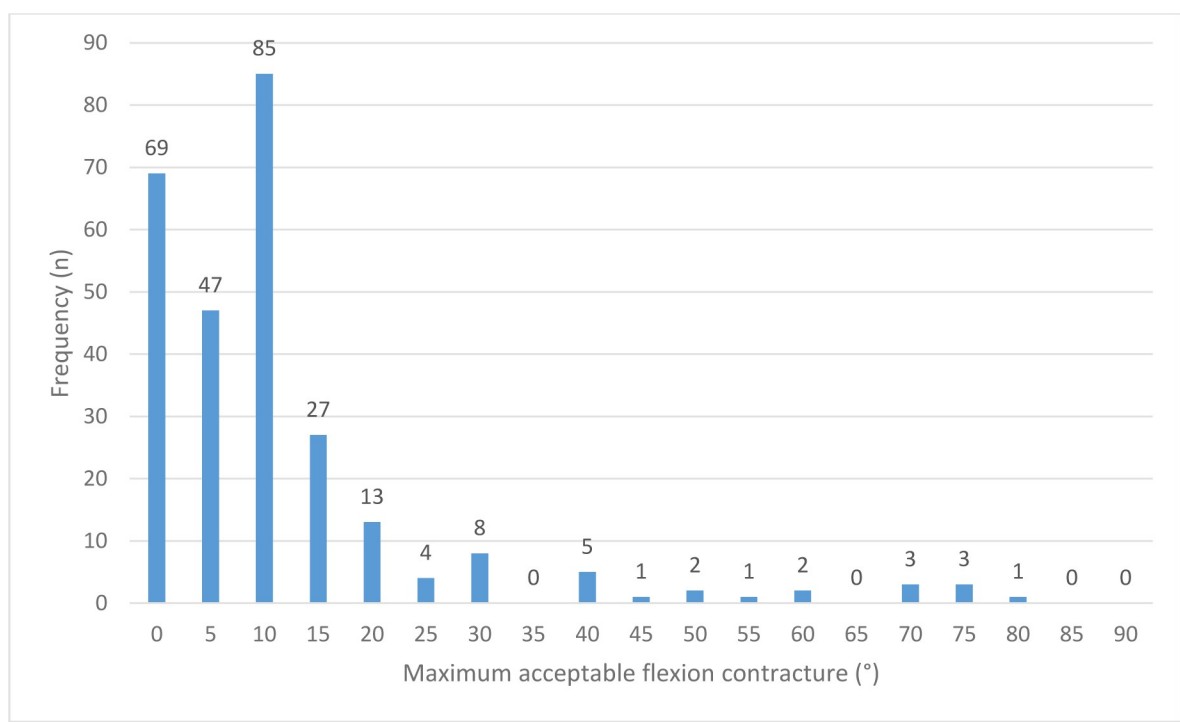

**Fig 2. Diagram of acceptable maximum contractures.**

rates for a modern BCR TKA after 18 months follow up,[13] this finding could not be supported by Baumann et al. and Alnachoukati et al. for the same implant.[2, 9, 14] Notably, older implants show good long-term survival up to 24 years.[4, 5]

There is a significant lack of knowledge around patient segmentation for BCR TKA. Most publications on BCR TKA do not specify limitations to indications.[2, 4, 5, 9, 13, 15]

Interestingly, the answers received indicated that BCR TKA could be an alternative to PKA. Possible reasons might be the combination of expected advantages of both procedures: the

**Table 6. Indication deformity.**

| Degree | Varus | | Valgus | | Slope | |
|---|---|---|---|---|---|---|
| ° | N | % | N | % | N | % |
| 0 | 17 | 6.3 | 32 | 11.9 | 13 | 4.9 |
| 5 | 51 | 19.0 | 63 | 23.5 | 84 | 31.3 |
| 10 | 115 | 42.9 | 113 | 42.2 | 132 | 49.3 |
| 15 | 46 | 17.2 | 43 | 16.0 | 29 | 10.8 |
| 20 | 26 | 9.7 | 10 | 3.7 | 5 | 1.9 |
| 25 | 6 | 2.2 | 2 | 0.8 | 2 | 0.8 |
| 30 | 5 | 1.9 | 4 | 1.5 | 1 | 0.4 |
| 35 | 2 | 0.8 | 1 | 0.4 | 0 | 0 |
| 40 | 0 | 0 | 0 | 0 | 1 | 0.4 |
| 45 | 0 | 0 | 0 | 0 | 1 | 0.4 |
| Total | 268 | 100% | 268 | 100% | 268 | 100% |
| Mean | 11.1° | | 9.3° | | 9.1° | |

78 surgeons did not answer this question.

**Table 7. Indication summary vs. BCR indication and PCL sacrifice.**

| | Do you sacrifice the PCL during TKA procedures? | | | | | | | | | | | | | | | |
| | Always | | | | | Sometimes | | | | | Never | | | | | |
| | N | Mean | Median | SD | Range | N | Mean | Median | SD | Range | N | Mean | Median | SD | Range | p-value[1] |
| Age | 70 | 66.7 | 65 | 10.50 | 36–99 | 62 | 69.0 | 70 | 9.66 | 47–90 | 14 | 68.3 | 70 | 8.14 | 55–82 | 0.3198 |
| BMI | 70 | 32.1 | 32 | 4.34 | 25–40 | 77 | 32.4 | 33 | 4.18 | 25–40 | 15 | 35.5 | 35 | 4.03 | 28–40 | 0.0274 |
| Contracture | 101 | 11.8 | 10 | 14.79 | 0–80 | 112 | 12.6 | 10 | 16.18 | 0–75 | 24 | 9.4 | 5 | 15.42 | 0–75 | 0.4693 |
| Varus | 88 | 12.4 | 10 | 6.21 | 5–30 | 111 | 11.7 | 10 | 5.37 | 5–30 | 17 | 12.6 | 10 | 7.93 | 5–35 | 0.8079 |
| Valgus | 84 | 11.1 | 10 | 5.17 | 5–30 | 106 | 10.6 | 10 | 5.29 | 5–35 | 15 | 12.3 | 10 | 6.23 | 5–30 | 0.4313 |
| Slope | 89 | 9.5 | 10 | 4.33 | 5–30 | 113 | 9.7 | 10 | 5.01 | 5–40 | 19 | 8.4 | 10 | 2.39 | 5–10 | 0.7226 |

[1] Kruskal Wallis test.

37 missing responses.

more natural knee kinematics achieved with UKA or bicompartmental arthroplasty and long-term survival of TKA. This might also be a reason for the higher patient satisfaction expected after BCR TKA. Johnson recorded as few as 7% of TKA patients reporting that their knee feels normal after surgery[6] while Alnachoukati et al. report 91% of BCR patients saying their knee always or sometimes feels normal.[14] Survival analysis of old BCR TKA designs with follow-up of up to 24 years reached 89% (82–93%) in a study by Pritchett. For modern designs, the follow-up times were shorter and results were heterogeneous. While Alnachoukati et al. reported 1.4% at 12 months,[16] Baumann et al. found revision rates of up to 5.9% at 18 months and,[9] Pelt et al. 17% revisions for any reason at 36 months for the same implant type.[17] Baumann et al. directly compared UKA, BCR TKA and standard TKA. They found BCR TKA and UKA to have comparable joint awareness in terms of the forgotten joint score (BCR TKA 53.4 vs. UKA 53.6 points). Both were superior to the standard TKA group (38.9 points) yet no objective knee scores were reported.[2, 9]

There is consensus regarding the importance of the ACL for natural knee kinematics and proprioception. A functionally intact ACL is a prerequisite for implantation of BCR TKA. In our study, surgeons reported an average of over 55% of patients undergoing TKA to have an intact ACL. This is supported by Johnson et al. and a current systematic literature review on the subject.[6, 18] Notably, Cloutier et al. did not find ACL degeneration to influence survival and clinical outcomes of BCR TKA in the long term.[19] In this survey, the majority of surgeons (90%) considered pre-interventional clinical function tests and 67% MRI to be to be reasonable.

**Table 8. Indication summary vs. BCR indication and general consideration of BCR TKA.**

| | Would you consider implanting a BCR total knee prosthesis? | | | | | | | | | | | | | | | |
| | Yes | | | | | No | | | | | Don't know | | | | | |
| | N | Mean | Median | SD | Range | N | Mean | Median | SD | Range | N | Mean | Median | SD | Range | p-value[1] |
| Age | 106 | 68.4 | 70 | 9.10 | 40–99 | 19 | 66.5 | 65 | 13.18 | 50–90 | 28 | 67.1 | 65.5 | 10.06 | 36–85 | 0.4202 |
| BMI | 127 | 32.9 | 34 | 4.13 | 25–40 | 21 | 31.7 | 30 | 4.70 | 25–40 | 29 | 31.6 | 31 | 4.39 | 25–40 | 0.2374 |
| Contracture | 189 | 13.2 | 10 | 15.89 | 0–80 | 36 | 4.6 | 0 | 5.90 | 0–20 | 46 | 10.1 | 10 | 12.27 | 0–75 | 0.0002 |
| Varus | 179 | 12.3 | 10 | 5.91 | 5–35 | 29 | 10.2 | 10 | 5.74 | 5–30 | 43 | 11.3 | 10 | 5.89 | 5–35 | 0.0813 |
| Valgus | 173 | 10.9 | 10 | 5.25 | 5–35 | 23 | 9.6 | 10 | 6.20 | 5–30 | 40 | 9.8 | 10 | 3.91 | 5–20 | 0.1535 |
| Slope | 181 | 9.4 | 10 | 3.86 | 5–25 | 31 | 9.7 | 10 | 5.31 | 5–30 | 42 | 9.4 | 10 | 5.97 | 5–40 | 0.7008 |

1: Kruskal Wallis test

While BCR TKA is often said to be developed for young and active patients, the literature shows wide ranges of age indicated for BCR TKA. In this survey, less than 50% of respondents considered patient age to be relevant. The mean upper age limit was set at around 68 years. While few surgeons would indicate BCR TKA in patients above 79 years of age, only about 9% would limit the indication to patients younger than 60 years. Overall, there was no clear trend toward excluding older patients.

It has been reported that BMI may be linked to complications and survival of TKA implants.[20] However, Baumann did not find a BMI influence on patient-reported outcomes and survival. In this survey, 47.5% of the knee surgeons considered high BMI to be a relevant factor for indication. While 18.7% limited BMI to pre-obesity, most would accept Class I and II obesity. However, only 10.8% would implant BCR TKA in patients with Class III obesity and 53.7% would not indicate above Class I. This is in line with current guidelines on TKA indications.[21]

Christensen et al. and Pritchett et al. restricted BCR TKA indication to minimal or limited flexion contractures.[13, 22] The limited possibility to address reasons for contracture during BCR TKA surgery may be the rationale for this. The majority of survey participants (n = 268) considered contractures relevant contraindications. 25.5% would not accept any contracture; 17.3% no more than 5˚ and 31.4% would accept up to 10˚. Only 15.9% accepted contractures of more than 15˚ (**Fig 2**).

Similar to contractures, the degree of deformity might limit the indication for BCR TKA. In most publications on BCR TKA, there were limitations to valgus deformity of maximum 10–30˚ and varus deformity of up to 10–30˚.[2, 15, 22] Underlying reasons may have been the assumption of impaired ACL function in severe deformities as well as impaired collateral ligament impairment. In this survey, restrictions to varus and valgus deformity were quite similar. Slightly less valgus deformity was accepted. The majority of surgeons would not indicate BCR TKA in more than 10˚ of varus (68.2%) or valgus (77.6%) deformity. These limitations may reflect the lack of constraint in BCR TKA which reduce the options of soft tissue balancing.

Interestingly, except for acceptable contractures, there were no differences regarding limitations between surgeons willing to perform BCR TKA or not. Similarly, PCL sacrifice preferences did not influence indication ranges. Only the upper BMI limit showed differences between the groups. However, previous experience with BCR TKA influenced the limitations regarding upper BMI, upper age, contractures and varus deformity. In all cases, experienced surgeons were less restrictive.

Of note, the number of patients considered eligible for BCR TKA showed significant differences between surgeons' PCL sacrifice preferences. Surgeons without strict PCL sacrifice regimes ("sometimes" vs. "always" or "never") were more likely to perform BCR TKA. This may indicate the openness of such surgeons to perform patient-specific procedures.

## Limitations

Firstly, the survey was distributed by an electronic mailing list that included non-knee surgeons. Therefore, the response rate from eligible candidates could not be calculated. However, the pre-study aim of at least 100 completed surveys for analysis was overachieved.

Secondly, most surgeons had no prior experience with BCR TKA. Therefore, this study represents opinions and assumptions. Still, these subjective opinions contain valuable information and help to better understand the potential role of BCR TKA in the spectrum of knee replacements. Additionally, the training and individual background of surgeons may affect answers. Specific differences between sports and joint fellowship trained physicians were not possible in this study.

Thirdly, the survey was only available in English and German. The perception towards BCR TKA might vary between cultures and multi-language surveys might be able to detect such variances.

However, despite these limitations, this survey is the first large scale sampling of surgeons' opinions of bi-cruciate retaining surgery.

## Conclusions

There is strong interest from orthopedic surgeons to perform BCR TKA. About 20% of their TKA patients may be eligible for BCR TKA. The survey showed the majority (75% or more) of surgeons would perform BCR TKA in patients up to an age of 80 years and with a deformity of up to 10˚ varus or valgus. Flexion contractures of up to 10˚ are acceptable to most of the survey respondents. BCR TKA for a patient with a BMI above 34.9 kg/m$^2$ is supported by few surgeons. BCR TKA experienced surgeons are less restrictive.

## Supporting information

**S1 File. English and German survey.** Complete questionnaire EXCEL file.
(XLSX)

**S2 File. Experience of participating surgeons.** Supplemental table.
(DOCX)

**S3 File. ACL intactness and suitability for BCR implantation.** Supplemental table.
(DOCX)

**S4 File. ACL status.**
(DOCX)

**S5 File. Individual answers.** Excel file with complete dataset.
(XLSX)

## Author Contributions

**Conceptualization:** Diarmuid De Faoite, Christoph Kolja Boese.

**Data curation:** Christoph Kolja Boese.

**Formal analysis:** Michelle Foster.

**Funding acquisition:** Christoph Kolja Boese.

**Investigation:** Christoph Kolja Boese.

**Methodology:** Christoph Kolja Boese.

**Project administration:** Christoph Kolja Boese.

**Writing – original draft:** Diarmuid De Faoite, Christoph Kolja Boese.

**Writing – review & editing:** Diarmuid De Faoite, Christian Ries, Michelle Foster, Christoph Kolja Boese.

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
