## [Decision Letter · Decision Letter 0]

13 Apr 2020

PONE-D-19-34268

Indications for bi-cruciate retaining total knee replacement: an international survey of 346 knee surgeons

PLOS ONE

Dear Diarmuid De Faoite,

Thank you for submitting your manuscript to PLOS ONE. After careful consideration, we feel that it has merit but does not fully meet PLOS ONE’s publication criteria as it currently stands. Therefore, we invite you to submit a revised version of the manuscript that addresses the points raised during the review process.

The reviewers' expressed some concerns about the methodology used in the current study. Also, they believe the authors need to discuss in detail about the indication and functional outcome of the modern BCR TKA compared to the traditional TKA.

For more information, please refer to the reviews' comments below.

We would appreciate receiving your revised manuscript by 6-11-2020. To enhance the reproducibility of your results, we recommend that if applicable you deposit your laboratory protocols in protocols.io, where a protocol can be assigned its own identifier (DOI) such that it can be cited independently in the future. For instructions see: http://journals.plos.org/plosone/s/submission-guidelines#loc-laboratory-protocols

We look forward to receiving your revised manuscript.

Kind regards,

Yifei Wang

Academic Editor

PLOS ONE

Journal Requirements:

2. Please include additional information regarding the survey or questionnaire used in the study and ensure that you have provided sufficient details that others could replicate the analyses. For instance, if you developed a questionnaire as part of this study and it is not under a copyright more restrictive than CC-BY, please include a copy, in both the original language and English, as Supporting Information."

3. Please provide additional details regarding participant consent. In the ethics statement in the Methods and online submission information, please ensure that you have specified (1) whether consent was informed and (2) what type you obtained (for instance, written or verbal, and if verbal, how it was documented and witnessed). If the need for consent was waived by the ethics committee, please include this information.

4. Thank you for stating the following in the Financial Disclosure section:"Funding was secured by CKB from investigator-initiated study funding provided by Smith + Nephew. (IIS #585).Website: https://www.smith-nephew.com/education/iis/The funders had no role in study design, data collection and analysis, decision to publish, or preparation of the manuscript."

Thank you for stating the following in the Competing Interests section:"I have read the journal's policy and the authors of this manuscript have the following competing interests: CKB, DDF and MS are employed by Smith & Nephew."

We note that one or more of the authors have an affiliation to the commercial funders of this research study : "Smith + Nephew"

b). Please also provide an updated Competing Interests Statement declaring this commercial affiliation along with any other relevant declarations relating to employment, consultancy, patents, products in development, or marketed products, etc. 

Reviewers' comments:

Reviewer's Responses to Questions

**Comments to the Author**

1. Is the manuscript technically sound, and do the data support the conclusions?

Reviewer #1: Yes

Reviewer #2: Yes

2. Has the statistical analysis been performed appropriately and rigorously? 

Reviewer #1: Yes

Reviewer #2: Yes

3. Have the authors made all data underlying the findings in their manuscript fully available?

Reviewer #1: Yes

Reviewer #2: Yes

4. Is the manuscript presented in an intelligible fashion and written in standard English?

Reviewer #1: Yes

Reviewer #2: Yes

5. Review Comments to the Author

Reviewer #1: This is a very useful survey based report of surgeons' opinions toward BCR TKA. The survey went through a formation process and the statistical analysis is well done and analyzed in a way that helps understand not only the surgeons' responses but how their practices correlate with their opinions on BCR TKA. Cleary the authors are suggesting in the paper that surgeons may be willing to embrace BCR TKA with more information and practice. To that end, I have the following suggestions for improvement:

1) The introduction and discussion present citations for the concepts of knee kinematics and difficulty of procedure as they relate to the opinions presented in the study. However, many surgeons may not embrace BCR TKA because of the lack of difference in objective functional outcomes or implant survival making it worth embracing a new more difficult technique. In addition to the study cited in lines 286-288, the discussion should have a citation regarding the survival rates of BCR TKA, and in addition to the citation in 284-85 regarding the forgotten knee sensation, an additional note should be made about the objective functional outcomes of UKA/BCR TKA compared to traditional TKA

2) The surgeons' experience in number of cases is presented, but the training may be an important component of surgeons' opinions as well. Perhaps sports fellowship trained physicians would be more likely to embrace this technique than joints fellowship trained physicians.

3) In the sections regarding an intact ACL (290-296), how was this determined? MRI, clinical exam, evaluation during surgery, or combination of the 3? Clearly the decision should be made prior to surgery if one is choosing to perform BCR TKA so it would be useful to have an objective measure of how this was determined

4) Given that the findings of this study are based on a theoretical adaptation of a surgical technique by a group of surgeons, was there any follow up to see if some actually embraced the concept and performed BCR TKA after the survey? Particular attention, or possible future study should look at surgeons who were not performing BCR TKA who then began performing the procedure, and evaluating why they switched, as well as comparing matched results from their patients undergoing the different procedures. If this can't be performed and included here, a comment on plans to perform that study may strengthen the impact of the current study.

5) In the section discussing contracture/deformity, reasons suggesting why one would use a border for too much deformity are presented. An additional possibility is that deformity beyond 10 degrees in either direction likely requires soft tissue release(s) for knee balancing that will require additional options for constraint not presented when performing BCR TKA. Perhaps a line commenting on the lack of constraint options is required here.

6) A section is dedicated to discussing ACL competence. Is there a reported ACL rupture rate requiring re-operation following BCR TKA that could be included here as surgeons weigh the possibility of switching to this procedure?

Overall this is a useful snapshot assessing how surgeons may approach whether to adapt a new procedure, with appropriate references for the reasoning behind adapting the new procedure presented in a coherent manner, strengthened by the acknowledged limitation that this is a survey based study of surgeons' opinions rather than a presentation of a fact-based argument.

Reviewer #2: General comments:

The authors have surveyed TKA surgeons including those who currently perform bicruicate raining TKA and those who do not, regarding their indications for TKA which retains both cruciate ligaments. Surgeons who perform UKA or CR TKA were found to more likely indicate BCR than surgeons who routinely sacrifice both cruciate ligaments during TKA. The study also found some consensus among surgeons regarding appropriate indications for the procedure. Bicruciate retaining TKA has been done in the past and was abandoned largely resulting from mechanical failure of the tibial construct. It would be helpful to compare the indications for modern bicruciate TKA’s in the current study to those used in the past since more narrow indications may affect the durability of the procedure.

Specific comments:

Introduction, last paragraph. What was the study hypothesis?

Materials and Methods, first paragraph. What were the fundamental areas of questioning included in the survey such as age, BMI, deformity, gender, and activity level, and why were they considered important? Why were two languages (English and German used for the survey)?

Materials and Methods, Survey design. The specific survey questions or most relevant ones should be included in a table or appendix.

Results. Were there any significant differences in reposes between the German and English speaking surgeons?

Results. Were there any significant differences in indications for surgeons who performed bicruciate retaining TKA and those without bicruciate retaining TKA experience?

Discussion, page 16. Are there any differences in surgeons’ indications for currently available modern bicruciate retaining TKA’s compared with those that had been used in the past and were abandoned?

Discussion, line 313. Some prior studies of earlier bicruciate retaining TKA’s (Pritchett, Cloutier) showered good survivorship. Were the indications used in prior studies of successful BCR similar or different than the authors current survey findings?

Tables. Tables 7 and 8 appear to contain information which is summarized adequately in the text and should be removed.

6. PLOS authors have the option to publish the peer review history of their article (what does this mean?). If published, this will include your full peer review and any attached files.

Reviewer #1: No

Reviewer #2: No

---

## [Author Response · Author response to Decision Letter 0]

12 May 2020

Please find the answers in the submitted "Rebuttal Letter". Thank you for reviewing our manuscript.

Yours sincerely,

The Authors

---

## [Decision Letter · Decision Letter 1]

1 Jun 2020

Indications for bi-cruciate retaining total knee replacement: an international survey of 346 knee surgeons

PONE-D-19-34268R1

Dear Dr. Diarmuid De Faoite,

We are pleased to inform you that your manuscript has been judged scientifically suitable for publication and will be formally accepted for publication once it complies with all outstanding technical requirements.

With kind regards,

Yifei Wang

Academic Editor

PLOS ONE

Additional Editor Comments (optional):

Reviewers' comments:

Reviewer's Responses to Questions

**Comments to the Author**

1. If the authors have adequately addressed your comments raised in a previous round of review and you feel that this manuscript is now acceptable for publication, you may indicate that here to bypass the “Comments to the Author” section, enter your conflict of interest statement in the “Confidential to Editor” section, and submit your "Accept" recommendation.

Reviewer #2: All comments have been addressed

2. Is the manuscript technically sound, and do the data support the conclusions?

Reviewer #2: Yes

3. Has the statistical analysis been performed appropriately and rigorously? 

Reviewer #2: Yes

4. Have the authors made all data underlying the findings in their manuscript fully available?

Reviewer #2: Yes

5. Is the manuscript presented in an intelligible fashion and written in standard English?

Reviewer #2: Yes

6. Review Comments to the Author

Reviewer #2: (No Response)

7. PLOS authors have the option to publish the peer review history of their article (what does this mean?). If published, this will include your full peer review and any attached files.

Reviewer #2: No

---

## [Editor Report · Acceptance letter]

4 Jun 2020

PONE-D-19-34268R1 

Indications for bi-cruciate retaining total knee replacement: an international survey of 346 knee surgeons 

Dear Dr. De Faoite:

I'm pleased to inform you that your manuscript has been deemed suitable for publication in PLOS ONE. Congratulations! Your manuscript is now with our production department. 

Kind regards, 

on behalf of

Dr. Yifei Wang 

Academic Editor

PLOS ONE